# Structure and Formation Mechanism of Antimicrobial Peptides Temporin B- and L-Induced Tubular Membrane Protrusion

**DOI:** 10.3390/ijms222011015

**Published:** 2021-10-13

**Authors:** Shan Zhang, Ming Ma, Zhuang Shao, Jincheng Zhang, Lei Fu, Xiangyuan Li, Weihai Fang, Lianghui Gao

**Affiliations:** Key Laboratory of Theoretical and Computational Photochemistry, Ministry of Education, College of Chemistry, Beijing Normal University, 19 Xin-Jie-Kou-Wai Street, Beijing 100875, China; shanzhang@mail.bnu.edu.cn (S.Z.); 202121150065@mail.bnu.edu.cn (M.M.); zshao0801@mail.bnu.edu.cn (Z.S.); 201711150185@mail.bnu.edu.cn (J.Z.); 201731150043@mail.bnu.edu.cn (L.F.); xyli@mail.bnu.edu.cn (X.L.); fangwh@bnu.edu.cn (W.F.)

**Keywords:** antimicrobial peptide, temporin B, temporin L, lipid membrane, molecular dynamics simulation

## Abstract

Temporins are a family of antimicrobial peptides (AMPs) isolated from frog skin, which are very short, weakly charged, and highly hydrophobic. They execute bactericidal activities in different ways from many other AMPs. This work investigated morphological changes of planar bilayer membranes composed of mixed zwitterionic and anionic phospholipids induced by temporin B and L (TB and TL) using all-atom and coarse-grained molecular dynamics simulations. We found that TB and TL fold to α-helices at the membrane surface and penetrate shallowly into the bilayer. These short AMPs have low propensity to induce membrane pore formation but possess high ability to extract lipids out. At relatively high peptide concentrations, the strong hydrophobicity of TB and TL promotes them to aggregate into clusters on the membrane surface. These aggregates attract a large amount of lipids out of the membrane to release compression induced by other dispersed peptides binding to the membrane. The extruded lipids mix evenly with the peptides in the cluster and form tubule-like protrusions. Certain water molecules follow the movement of lipids, which not only fill the cavities of the protrusion but also assist in maintaining the tubular structures. In contrast, the peptide-free leaflet remains intact. The present results unravel distinctive antimicrobial mechanisms of temporins disturbing membranes.

## 1. Introduction

Antimicrobial peptides (AMPs) widely exist in animals and plants, which are the key components of the innate immune system. AMPs display a broad spectrum of antibacterial activity against Gram-negative and Gram-positive bacteria, fungi, yeast and viruses [1,2,3]. In addition, they have a low propensity to induce drug resistance, thus provide the possibility to develop a new class of antibiotics [1,4,5]. The core of AMPs’ antibacterial activity is their interaction with bacterial plasma membrane. Most AMPs are small, positively charged, and amphiphilic. The cationic residues provide an electrostatic driving force for the binding of AMPs with negatively charged bacterial membrane. Then, their small hydrophobic motifs tend to insert into the lipid head groups and disturb the membrane via formation of membrane pores or membrane morphology changes [6,7]. A series of hole models have been proposed. In a toroidal pore model, a couple of AMP molecules are supposed to associate with lipid heads to form a transmembrane water-permeable pore [8,9]. In a barrel-stave model, only peptides accumulate into barrel shaped aggregates, also showing transmembrane orientation and water-permeable pores [10,11,12]. A carpet model provides an alternative view where AMP are in contact with the lipid head peptides and spread on the surfaces of the membrane, covering the membrane like a carpet. In this model, when the peptide concentration reaches a critical value, the membrane collapses to form defects and dissolves into micelles [12,13,14].

Temporins represent a family of cationic antimicrobial peptides, which were first isolated from skin secretions of the European red frog Rana temporaria [15]. Temporins show special structural properties. They contain only 10–14 residues, making them one of the shortest antimicrobial peptides. At the same time, they have few positive charges, from 0 to +4 at physiological pH conditions. Nevertheless, they can change conformations from random coil to α-helical and β-sheet structures depending on their solvent environments and cause significant perturbations of the membrane [16]. For example, temporin B (LLPIVGNLLKSLL), only having one positively charged residue, is active against Gram-positive bacteria and fungi, and basically shows no hemolysis and toxicity to normal human cells [17,18,19,20]. Temporin L (FVQWFSKFLGRIL), rich in aromatic residues and containing two positively charged amino acids, has broad-spectrum antibacterial activity and is effective against both Gram-positive and Gram-negative bacteria, fungi, and cancer cells. However, TL is lysis to erythrocytes [18,21,22,23]. These properties make temporins good templates for synthesizing new antimicrobial peptide drugs.

Many experimental investigations have been conducted on TB and TL in relation to membranes. However, their antimicrobial mechanisms still remain ambiguous and debated. Saviello et al. suggested “carpet-like” or “dynamic peptide-lipid supramolecular pore” models, which do not need any particular peptide conformation, to explain temporins’ antimicrobial activity, but a tail-to-tail dimerization barrel-stave model to interpret the hemolytic effect of TL [24]. In contrast, permeabilization of temporin-treated lipid vesicles revealed a marked dependence of peptide-induced leakage on the molecular mass/size of the fluorescent probe, which indicated that these peptides do not have a detergent-like effect on the membrane, suggesting that perturbation of bilayer organization takes place on a local scale through a formation of membrane pore [19]. Therefore, Kinnunen et al. suggested a different folding/aggregation model, which supposed that α-helical peptides associate into oligomers and form protofibril which permeates membranes and triggers cell death via “leaky slit” [25,26]. Interestingly, it was also found that TB and TL can induce rapid vesiculation of giant unilamellar vesicles (GUV) when anionic lipid was present [18] and formation of tubular lipid protrusions from supported phospholipid bilayers, but the mechanistic basis for these features of the vesicle budding and tubules are uncertain [27]. 

Commonly used structure detections by optical techniques have some obstacles when applied to peptide–membrane systems. The details and dynamic processes of the interactions between peptides and membranes cannot be measured directly. As auxiliary methods, molecular dynamics simulations provide molecular and atomic structural details of peptides and membranes as well as their dynamic changes. All-atom (AA) molecular dynamics (MD) simulations have been adopted to investigate the interactions of various AMPs with model membranes, such as melittin [28], magainin [29], HIV-1 tat [30] and cyclic peptide [31]. The mechanisms of pore formation and micellar decomposition were proposed through simulations. In addition, coarse-grained (CG) molecular simulations are also widely used to simulate large peptide–membrane systems for a long time, which improves the time and space limitation of all-atom simulations. In recent years, we have used CGMD to investigate the interactions of several AMPs with lipid bilayers and vesicles, including α-helical magainin, melittin, and CM15, β-sheet protegrin-1, tachyplesin I, and gomesin, extended HIV-1 Tat and indolicidin, as well as cyclic polymyxin B [32,33,34,35,36,37,38,39]. We found that peptides with different structures and residue compositions destroy the membrane in a variety of ways. In this work, both AA and CG molecular dynamics simulations were employed to study the activities of temporins acting on bacterial membrane models. More specifically, we investigated the secondary structures of TB and TL on the water–gas interface and membrane surface, the stability of peptide dimers and oligomers, as well as their influences on the membrane integrity, morphology, and tensile kinetics. Our results clearly elucidated the antimicrobial mechanisms of temporins and their special manners disturbing membranes. 

## 2. Materials and Methods

### 2.1. All-Atom Simulation Parameters and Protocol

Peptides at water–gas interface and lipid bilayer surface were simulated by all-atom molecular dynamics using GROMACS 2018.8 suite [40,41] with GROMOS 54a7 force field [42] for peptides and GROMOS 53a6 force fields for lipids [43]. The initial structure of temporin B was obtained from the nuclear magnetic resonance (NMR) structure in lipopolysaccharide micelles [44], and temporin L was taken from PDB bank with ID: 6GS5 [45]. According to the determination of NMR, the initial conformations of both peptides were set to α-helices. 

For the water–gas system, a box with an initial size of 6.0 × 6.0 × 10.0 nm^3^ was prepared. Peptide monomer or dimer were placed near the water–gas interface with their helical axes parallel to the interface. Then, the half of the box containing peptides were solvated with simple point-charge (SPC) water model [46]. Counter ions Cl^−^ were randomly added to neutralize the system. After energy minimization, MD simulations were performed in constant number, volume, and temperature (NVT) ensemble. Temperature was set at 310 K as default. Energy minimization was carried out using steepest descent algorithm until the maximum force was less than 1000.0 kJ/mol/nm. Relaxation of solvent was first carried out for 100 ps by restricting the positions of peptides. Then, product simulations were conducted in the NVT ensemble for 100 ns with periodic boundary conditions with a time step of 2 fs. Bond lengths were constrained with the LINCS algorithm [47]. Long-range electrostatic interactions were calculated by applying Particle-mesh Ewald (PME) algorithm [48]. The cutoff radius of both Van der Waals and Coulomb interactions was 1.2 nm. 

For the peptide–membrane systems, the lipid bilayer was a mixture of zwitterionic palmitoyloleoyl phosphatidylcholine (POPC) and anionic palmitoyloleoyl phosphatidylglycerol (POPG) lipids with molar ratio 7:3 to mimic bacterial plasma membrane as in many experiments [18,21,25,26,27]. Real bacterial plasma membrane also contains a small amount of cardiolipins. This kind of anionic lipid serves as a binding target of AMPs and may modulate membrane bending rigidity. For simplicity, cardiolipin molecules were not incorporated in the model membranes in our simulations. We speculate that it may not significantly affect the interaction mechanism between membrane and AMPs. Specifically, a bilayer containing a total of 200 lipids (140 POPC and 60 POPG molecules) with a size of 8.3 × 8.3 nm^2^ was prepared in advance and placed in the center of box of 8.3 × 8.3 × 13.0 nm^3^. Peptide dimers at various concentration were placed about 1–2 nm above the upper leaflet of the phospholipid bilayer. The initial structures of the peptide dimer were taken from the water–gas interface simulations. Their axes were set parallel to the membrane plane. The bilayer and peptides were solvated by SPC water. A certain amount of Na^+^ and Cl^−^ ions were added to neutralize the solution, and the concentration of NaCl was 0.1 mol/L. After energy minimization, 100 ps simulations in NVT ensemble followed by 1000 ps simulations in NPT (constant pressure) ensemble with position restraints on the peptide and lipid atoms were carried out to equilibrate solvent. Product simulations (no restraints) were run for 200 ns in the NPT ensemble. The pressure was kept constant (1 bar) using semi-isotropic Parrinello-Rahman barostat with coupling time of 2.0 ps [49]. Other simulation settings were the same to the water–gas interface system as described above. Two independent samples were simulated for each system set. Analyses were conducted on the trajectory of the last 50 ns of each simulation.

### 2.2. Coarse-Gained Simulation

In order to investigate the interactions between peptides and membranes for a long time and large size scales, bigger peptide–membrane systems were simulated using Gromacs 5.0.4 software (http://www.gromacs.org/, accessed on 11 May 2021) with MARTINI 2.2 coarse-grained force field. The CG models of lipids and peptides were established based on the four-to-one mapping scheme of Martini model (Figure 1) [50,51]. Two systems of different sizes were implemented: a small bilayer patch composed of 200 lipids bound by one peptide monomer in a box with an initial size of 8.0 × 8.0 × 14 nm^3^ was used to compare with AAMD results; a big bilayer membrane composed of 1200 lipids interacting with peptide with peptide/lipid (*P/L*) ratio ranging from 1% to 10% in a box of 19.9 × 19.3 × 18 nm^3^ was used to record possible global morphological changes. The lipid compositions of each bilayer were the same as that of the AA simulations (POPC:POPG = 7:3). After placing peptides on one side of the membrane surface, Na^+^ ions were added to neutralize the negative charge in the membrane, and Cl^−^ ions were added to neutralize the positive charges of peptides. All production were simulated in NPT ensemble with temperature at 310 K. Parrinello-Rahman barostat maintained the pressure at 1.0 bar with semi-isotropic coupling [49]. All CG simulations were carried out at time step of 0.02 ps under periodic boundary conditions. For each system, up to 1 μs simulation was performed and repeated for three times. Then, 1000 samples in the last 200 ns of each trajectory were selected for quantitative analysis. System configurations were visualized by VMD software [52].

### 2.3. Simulation Analyses

#### 2.3.1. Potential of Mean Force

The Potential of Mean Force (PMF) for two TB or TL monomers to combine into a dimer at water–gas interface was determined by umbrella sampling [53]. The initial structures of the peptides were in perfect α-helical conformations. The two peptides were placed in reverse parallel at the water–gas interface, with their helical axis in parallel to the interface plane. Forty-one windows were prepared, in which the centroids of the two peptides were confined within a certain distance from 0.0 nm to 4.0 nm per 0.1 nm by a constant harmonic force of 1000 kJ/mol/nm^2^. After pre-equilibration, each window was simulated for 30 ns at 310 K, and the trajectory in the last 10 ns was utilized to produce PMF profiles by weighted histogram analysis (WHAM) method [54] using g_wham GROMACS tool. 

#### 2.3.2. Lipid Orientation Order

The adsorption of AMPs on a membrane crowds surrounding phospholipids resulting in global lipid rearrangement. The orientation order of lipid tail, $S_{\mathrm{tail}}$, can well reflect this, which was calculated as follows:(1)$$S_{\mathrm{tail}}=0.5\;\left({3\langle \cos}^{2}\varphi \rangle -1\right).$$

Here, *φ* was the angle between the normal of the bilayer plane and the lipid hydrocarbon tail vector. Ensemble average was performed over all lipids and trajectory samples.

#### 2.3.3. Membrane Bending Rigidity

Bending rigidity is a significant thermodynamic property of membranes, which can be obtained by fitting the orientation fluctuation spectrum of lipids [55,56]. In this method, the bilayer was mapped to a coarser discrete grid with spacing *h* of about 1.3 nm. In each mesh, the lipid tail orientation vectors $n_{j}^{\alpha }$ (*α* = 1 and 2 represent outer and inner monolayer, *j* is the *j*th tail) were first projected onto the *x*-*y* plane, then mapped onto a two-dimensional real space grid. The average projection of all vectors on current mesh $\left(m_{x},m_{y}\right)$ was defined as a transverse vector $n^{\alpha }\left(r\right)$. Here r=hm, $m=m_{x}i+m_{y}j$. The Fourier transformation of the lipid orientation vector $n\left(r\right)=\frac{1}{2}\left[n^{1}\left(r\right)-n^{2}\left(r\right)\right]$ yielded:(2)$${\hat{n}}_{q}=\frac{h^{2}}{L_{x}L_{y}}\underset{m}{\sum }n\left(r\right)e^{-iq\cdot \;r}.$$

Here, $q=\left(\frac{2\pi m_{x}}{L_{x}},\frac{2\pi m_{y}}{L_{y}}\right)$ with $-\frac{L_{x}}{2h}\leq m_{x}\leq \frac{L_{y}}{2h}$ and $-\frac{L_{x}}{2h}\leq m_{y}\leq \frac{L_{y}}{2h}$. Then, thermal fluctuation of the lipid orientation, $S_{u}\left(q\right)$, was obtained:(3)$$S_{u}\left(q\right)=L_{x}L_{y}{\langle |{\hat{n}}_{q}^{∥}|}^{2}\rangle =\frac{k_{B}T}{\kappa q^{2}}.$$

Here, ${\hat{n}}_{q}^{∥}=\left[q\cdot {\hat{n}}_{q}\right]/q$ was the longitudinal component of ${\hat{n}}_{q}$, $k_{B}$ was the Boltzmann constant, and *q* was the wave vector. The bending modulus κ can be obtained by fitting Equation (3). Converged values of κ were obtained from the plateau regions. To calculate κ of AMP-treated membrane, the same amount of TB or TL should be symmetrically adsorbed to the upper and lower leaflets of the bilayer.

#### 2.3.4. Lipid Packing Defect

Penetration and insertion of AMPs on a membrane perturb lipid arrangement and lateral packing pressure distribution, which may lead to defects in the membrane. Chemical defects occur when solvent accesses alkane chain. If the peptide-induced voids are deeper than a glycerol group, they are defined as geometrical defects [57,58]. To estimate the sizes of the defect and their distributions, the membrane was first divided into grids with a side length of 0.1 nm, thus an elementary defect had an area of 0.01 nm^2^. Then, the lipid beads were detected vertically from the outer leaflet bound by peptides. The results fell into three categories: (i) If the first bead detected was a polarity head bead of a lipid, the mesh was discarded. (ii) If it was a hydrocarbon bead, it was defined as a chemical defect. (iii) If the first detected bead was a hydrocarbon bead and its vertical position was lower than that of the nearest glycerol bead, the mesh was considered as a geometrical defect. Adjacent elementary defects merged to a larger defect.

## 3. Results

### 3.1. Temporin B and L Form Stable Helical Dimers at Water–Gas Interface

TB and TL have no stable secondary structure in aqueous solution. However, at the water–gas interface, they fold to well-ordered helical conformations (see Appendix A for the helicity, gyration radius, and solvent accessible surface area). The water–gas interface provides a stable amphiphilic environment, which promotes the hydrophilic and hydrophobic residues of peptides to take different orientations: the hydrophilic residues face the water layer, and the hydrophobic residues face the gas layer. We also placed two TBs or TLs at the water–gas interface in an antiparallel manner to investigate their capacities to form dimers. The dimerization free energy in Appendix A shows that they tend to aggregate to a stable helical dimer at distance of 1.0 nm. Moreover, two TLs are more likely to form a dimer structure and have higher helicities than TBs (Appendix A).

### 3.2. Temporin B and L Maintain α-Helix Conformations at Lipid Membrane Surface

To mimic the interactions between temporins and bacterial membrane, we built a bilayer patch composed of 200 phospholipids with POPC:POPG = 7:3 in AAMD simulations. One to six TB or TL dimers were placed on one of the bilayer surfaces in grid, respectively. The initial structures of the peptides are the stable helical dimers obtained from the preceding water–gas system.

TB and TL both adopt two poses when they bind onto the membrane surface, as shown in Figure 2. One is an inclined pose, where the positively charged N-terminus of the peptide is electrostatically adsorbed onto the negatively charged membrane, while the C-terminal section inclines upward in the aqueous solution. Because the N-terminal section of both TB and TL has two hydrophobic residues, it can embed into the hydrophobic core of the bilayer, especially at higher peptide concentration. In another pose, the helical axes of the peptides are in parallel to the surface of the membrane, and all positively charged residues are electrostatically adsorbed onto the membrane. TBs and TLs can basically maintain ordered α-helical structures at the membrane surface (see Appendix A). The helicity of TL is still higher than that of TB because the rich aromatic ring-containing residues (TRP^4^, PHE^5^, PHE^8^) in TL can stabilize its secondary structure at the amphiphilic membrane–water interface. However, compared to the water–gas interface, the gyration radii and solvent accessible surface areas of both peptides at membrane surface increase (Appendix A), while their helicity relatively decrease (Appendix A), indicating that the peptides partially unfold on the membrane surface.

### 3.3. Temporin B and L Squeeze out Lipids from Membrane

Upon binding to a lipid bilayer membrane, TB dimers at lower peptide concentration (*P/L* = 1% and 2%) dissociate and interact with the membrane via monomers. In contrast, TLs can basically retain the original dimer structures on the membrane surface at any peptide concentration (Appendix A). The adsorbed peptides perturb the lipid packing by penetration into the membrane via hydrophobic interaction such that the lipids underneath the peptides are pressed inward and the membrane becomes thinner. Meanwhile, the lipids surrounding the peptide are subjected to an extrusion pressure and become tilted greatly. Lipids close to the peptides are squeezed out of the membrane plane. Two-dimensional distribution maps of lipid heights (represented by phosphorous atoms) in the peptide-bound leaflet in Figure 3 illustrate these effects: At *P/L* > 2%, the relative height difference between the head groups is up to 2.6 nm, exceeding the thermal fluctuation amplitude of a membrane (around 1 nm). Lipid extrusion results in the increased membrane area, decreased membrane thickness, and enhanced lipid tail disordering (Figure 4A–C). Our simulations also demonstrated that TLs have relatively less ability to squeeze out lipids than TBs. This is because TLs tend to form dimers which effectively have smaller hydrophobic facets than the corresponding two monomers. In addition, the association of N- and C-termini in TL dimers lowers their interactions with lipids.

### 3.4. Temporin B and L Penetrate Shallowly into Membrane

Despite that TB and TL can disrupt the morphology of membrane by dragging out lipids, they cannot penetrate deeply into the membrane and translocate across it at peptide concentration up to *P/L* = 6%. Unlike many pore-forming peptides (such as melittin [28] and margainin [29], which have lengths comparable to the membrane thickness), the short length (less than half of the membrane thickness) of this kind of AMPs makes their vertical insertion states unfavorable. 

Figure 5A presents the snapshots of the stable bound states of TB and TL monomer on the membrane surface (simulated for 200 ns). The insertion depth (represented by the position of each Cα atom relative to the average position of phosphorous atom in the upper leaflet in the bilayer normal direction) given in Figure 5B well illustrates their shallow penetration. Most of the backbones stay in the lipid head region with the N-terminus sloping downward. Except the first two residues at N-terminus, TL inserts slightly deeper than TB.

Figure 5C,D also presents the snapshots and insertion depths of TB and TL monomers on membranes obtained from CG simulations. Here, the insertion depth was represented by the distance from a backbone bead to the average height of phosphorous beads in the upper leaflet. Similar shallow-penetration was found. Because angle and torsion constraints were applied to maintain the helical secondary structures of peptides in CG models, the peptides insert deeper and show better sequences-dependent patterns than those in AA models. The hydrophobic residues PRO^3^, ILE^4^, LEU^8,9^ of TB and PHE^5,8^, LEU^9^ of TL have relatively deep insertion depths, but are still all less than 1 nm. Based on these consistencies between AA and CG models, we then performed much longer CG simulations for larger systems to investigate the actions of AMPs on membrane in a more global view. 

### 3.5. Temporin B and L Induce Tubule-like Membrane Protrusions

In the CGMD simulations, temporins with concentrations *P/L* = 1–10% were placed on one surface of a membrane containing 1200 phospholipids with POPC:POPG = 7:3. The range of the peptide concentration covered the concentrations that can induce vesicle leakage (*P/L* > 0.2%) [21] and tubule on supported bilayer (*P/L* < 14%) [27]. Figure 6 gives the final snapshots simulated for 1 μs and Figure 7 shows the time sequences of the membrane morphologies induced by TB and TL at *P/L* = 10%. At low peptide concentrations (*P/L* = 1–5%), TBs and TLs are almost evenly adsorbed on the membrane surface. Though the membrane shows obvious fluctuation, the integrity of the planar bilayer maintains. When *P/L* exceeds 6%, the highly dense peptides tend to aggregate via hydrophobic interaction. At such high peptide concentrations, the electrostatic interactions between the AMPs and anionic lipids reach saturation such that the aggregated peptides cannot disperse evenly upon binding to the membrane surface but crowd into a peptide-rich region. These peptide clusters have low propensity to penetrate into the membrane but possess ability to release the strong compression induced by the adsorption of dispersed AMPs via dragging out surrounding phospholipids. The extruded lipids mix with the clustered peptides and form an arched bulge structure. Following the movement of lipids, water molecules continuously enter and enlarge the cavity between the bulge and the parent membrane. Tubular structures composed of both peptides and lipids emerge. Since the peptides only shallowly penetrate into the proximal leaflet, the distal leaflet remains intact in the process of tubule formation.

In a few simulated samples, the peptides were observed forming multiple small aggregates (for example TB at *P/L* = 7.3%). The clustered peptides coordinate and insert into the hydrophobic core region of the membrane forming barrel-like transmembrane channels. Hydrophilic residues crowd in the inner wall of the channel (invisible in Figure 8) and permit water to pass. The pores are metastable, and the peptides finally translocate to the opposite leaflet of the bilayer. We did not observe TL-induced pore formation.

We note that in order to accelerate the peptide binding but avoid improper binding onto the distal leaflet of a membrane in a periodic box, peptides were placed very close to the upper membrane surface (less than 2 nm) in our simulations. As a result, at high peptide concentrations, AMPs aggregate quickly before they approach to the membrane surface due to hydrophobic attractions (see the early binding process at *P/L* = 10% in Appendix A). If the peptide were placed more dispersed in the initial condition, multiple small aggregates may form. They may further fuse and bind to the membrane surface in a slow fashion. Nevertheless, we speculate that the results obtained so far will not change.

### 3.6. Temporin B and L Create Lipid Packing Defects

To fully understand the formation mechanism of the temporin-induced bulges and tubules, quantitative analyses of membrane properties are essential. We first estimated lipid packing defects induced by TB and TL at different peptide concentrations. Figure 9A,B shows that the emerging probability of chemical defect decreases exponentially as the size (or area, A) of the defect increases, which can be well fitted by $p\left(A\right)=B\times 10^{-kA}$, where k is the exponential decay rate (Figure 9C). Small k means that larger defects can be created. Results in Figure 9 demonstrate that the more AMPs are adsorbed to the membrane surface, the larger chemical defects are produced. The largest chemical defect size is around 2.5 nm^2^. On the other hand, the geometrical defect sizes are smaller than 1.2 nm^2^ and less dependent on the peptide concentration, indicating that the AMPs only shallowly bind on the membrane surface. Compared with TB, TL has one more electric charge which permits it to bind more closely with negatively charged POPG. Meanwhile, the three aromatic residues of TL are more likely to contact with the lipid alkane groups. These two features enable TL to create relatively larger chemical defects, which may explain its cytotoxicity. However, at *P/L* > 3%, TLs create smaller geometrical defects (indicated by larger k value) than TBs. We speculate that the propensity to aggregate into dimmer and oligomer of TLs reduces their penetration efficacy. 

### 3.7. Binding of TB and TL Enhances the Flexibility of a Membrane

The binding of TB and TL also changes the elasticities of the membrane, such as bending rigidity modulus κ (Figure 10). Here, we estimated κ by fitting the longitudinal lipid orientation fluctuation spectrum (Appendix A). When sufficient AMPs bind to the membrane, the membrane bending modulus κ decreases from 11.29 × 10^−20^ J (*P/L* = 0) to 9.41 × 10^−20^ J and 9.73 × 10^−20^ J (*P/L* = 5%) for TB and TL, respectively. The softening of the temporin-treated membrane facilitates formation of membrane tubule.

## 4. Discussion

Thousands of AMPs have been discovered and documented since the 1980s. They have high diversity in their sequence composition, length, and secondary structure and exercise antibiotic functions by damaging the bacterial membrane, translocating across the cell membrane to bind with intracellular DNA or RNA, or preventing intracellular synthesis. Among them, the almost shortest, weakly charged, and highly hydrophobic linear peptide temporins composed of conventional amino acids provide good guidelines for further modification and design of new pharmaceutical antimicrobial peptides. In this work, we systematically studied the interactions between temporin B and L and lipid bilayer composed of POPC and POPG by using AA and CG molecular dynamics simulations. Detailed antimicrobial mechanisms and pathways were elucidated. 

When exposed to amphipathic environments, such as water–gas interface and lipid bilayer surface, both TB and TL fold to α-helical conformations with their hydrophilic and hydrophobic residues separated into two facets. They favor to form CN-NC antiparallel dimer at the water–gas interface, while on the membrane surface, TBs separate and prefer monomer states but TLs can more or less stay in dimer structure. These are consistent with the previous experimental results revealed by circular dichroism and NMR techniques [17,44]. We speculate that the anionic heads of POPG lipids attract both the cationic residues and N-termini of the peptides, which destabilizes the dimer formation. Nevertheless, TL has one more charged residue than TB, which allows it to efficiently neutralize the nearby anionic heads of POPG lipids, leaving opportunity for the N- and C-termini to still interact via electrostatic attraction and retain dimer.

At relatively low peptide concentration, TBs and TLs spread on the membrane surface mainly interacting with the lipid head groups rather than deeply inserting into the bilayer. This is consistent with experimental research [17]. Despite that the binding of temporins induces phospholipid tilting, packing defects and membrane thinning, the integrity of the membrane maintains. Compared with many other α-helical AMPs, TB and TL are too short to span the membrane alone. 

At higher peptide concentration, temporins tend to aggregate through hydrophobic interaction when they are adsorbed onto a membrane and form peptide-rich regions on the membrane surface. The peptide clusters attract lipids to extrude out of the proximal leaflet of the membrane. The protruded lipids move on the surfaces of the peptide bundle, mix with them, and finally fuse to form arched tubule-like bulges. The cavity of the tubule is filled with water; meanwhile, the peptides distribute almost evenly on the inner and outer walls of the tubules. These mechanisms can well explain the observations of temporin-induced vesiculation of GUV and tubular lipid protrusions from supported phospholipid bilayers in experiments [18,27]. Synthesized peptide KLA1 (KLALKLALKAWKAALKLA-NH2) [59] was also found changing membrane morphologies in a similar manner. These indicate that lipid tubule formation and budding of vesicles might be general phenomena occurring after the binding of short amphipathic peptides to membranes. Our simulations here provided more detailed information about the structures and compositions of the protrusion, as well as the formation mechanism. 

A few events of dynamic pore formation were also observed for TB in our simulations. At peptide-rich regions, some N-termini can insert deeply and pull other peptides continuously to diffuse and translocate across the membrane. Transient membrane pores were present. A small amount of water was found passing through the channels. In the pore state, the orientations of lipids are not significantly different from those in a peptide-free membrane, indicating that the pore resembles the barrel model. However, our CG simulation cannot identify whether TBs form tail-to-tail dimers [24] or protofibril [25,26]. The arrangements of the peptides in a pore are sort of irregular. 

Comparing TB and TL, we found that TL possesses slightly higher antimicrobial activity than TB at low peptide concentration (*P/L* < 3%) as indicated by the deeper penetration (Figure 5) and smaller defect size induced by TL monomers (Figure 9). This effect is attributed to the three aromatic residues of TL that permit it to penetrate deeply into the membrane. However, at higher peptide concentration, TLs are more likely to associate into dimers, which on the contrary lowers their antibacterial activity as indicated by the reduced geometrical defect size (or larger k shown in Figure 9D) and rare membrane pore formation. Previous studies have shown that aggregation of AMPs may lower their potency to disrupt the membrane and reduce their antibacterial efficacy [60,61,62]. For example, Zai et al. recently showed that removal of Phe residue at N-terminus and substitution of Trp/D-Trp at C-terminus of a novel temporin peptide, Temporin-PF (FLPLIAGLFGKIF), resulted in an enhanced inter-peptide interaction in the zipper-like domain and eliminated their overall biological activities [62]. This implies that hydrophobicity and aggregation level of AMPs must be balanced to achieve maximum antimicrobial efficacy. 

## 5. Conclusions

In conclusion, we unraveled unique antimicrobial mechanisms of short peptides temporin B and L by combining AA and CG molecular dynamics simulations. In addition to creating dynamic permeable pores, these AMPs possess a special ability to drag out lipids from the membrane to form tubule-like protrusions. Our findings show that these short peptides are unfavorable for deep and stable insertion into membrane. Alternatively, their high hydrophobicity enables them readily to associate into clusters at high peptide concentration. These aggregates attract lipids out of the membrane to release compression induced by other dispersed peptides that reach the membrane in advance. The extruded lipids carry water, mix with the clustered peptides, and finally form water-filled tubular protrusions. The protrusions only occur on the peptide-rich leaflet of the membrane, while the peptide-free leaflet remains intact. These results shed distinct light on the activities of very short AMPs and provide ideas for design of novel antibiotic agents.

## Figures and Tables

**Figure 1 ijms-22-11015-f001:**
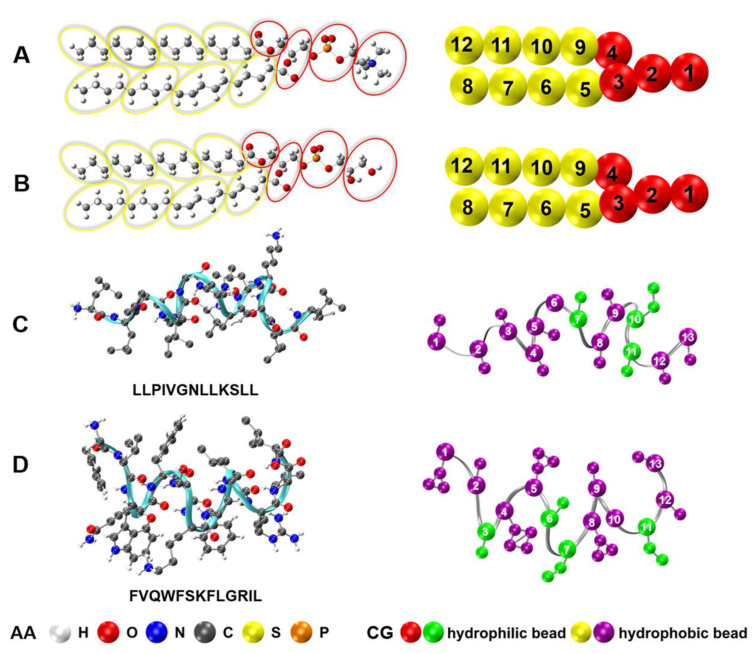
Schematic illustration of molecular structures of (**A**) POPC, (**B**) POPG, (**C**) temporin B and (**D**) temporin L, and their corresponding coarse-grained mappings. For POPC and POPG, the CG hydrophilic head beads are in red and the hydrophobic tail beads are in yellow. For TB and TL, the hydrophilic residues are in green and the hydrophobic residue beads are in purple.

**Figure 2 ijms-22-11015-f002:**
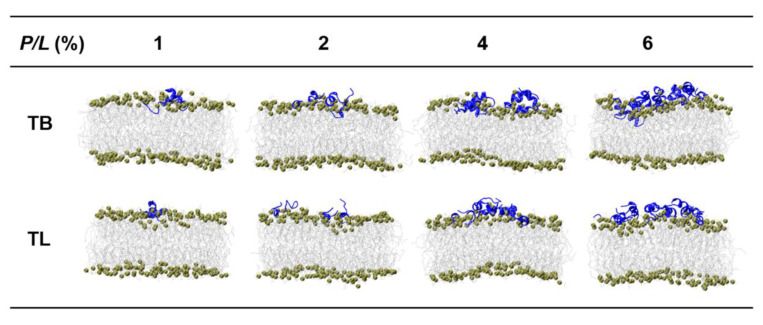
Snapshots of TBs and TLs at different peptide concentrations interacting with membrane. Blue ribbons represent peptides, tan beads represent phosphorous atoms, silver chains represent alkyl tails of lipid.

**Figure 3 ijms-22-11015-f003:**
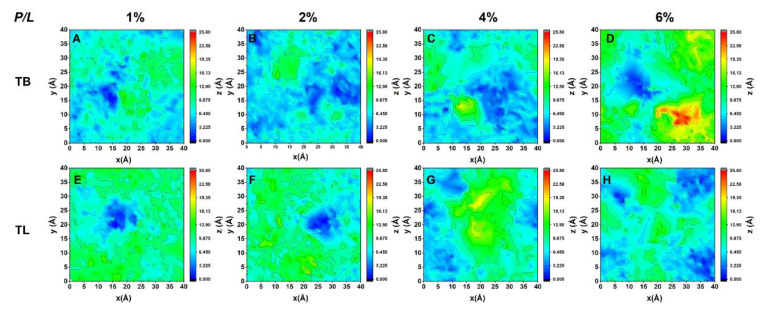
The 2D lipid height map of TB- and TL-treated bilayer membrane. The height of phospholipid was defined as the distance between a phosphorous atom in the peptide-bound leaflet relative to the lowest phosphorous atom in the same leaflet in the bilayer normal direction.

**Figure 4 ijms-22-11015-f004:**
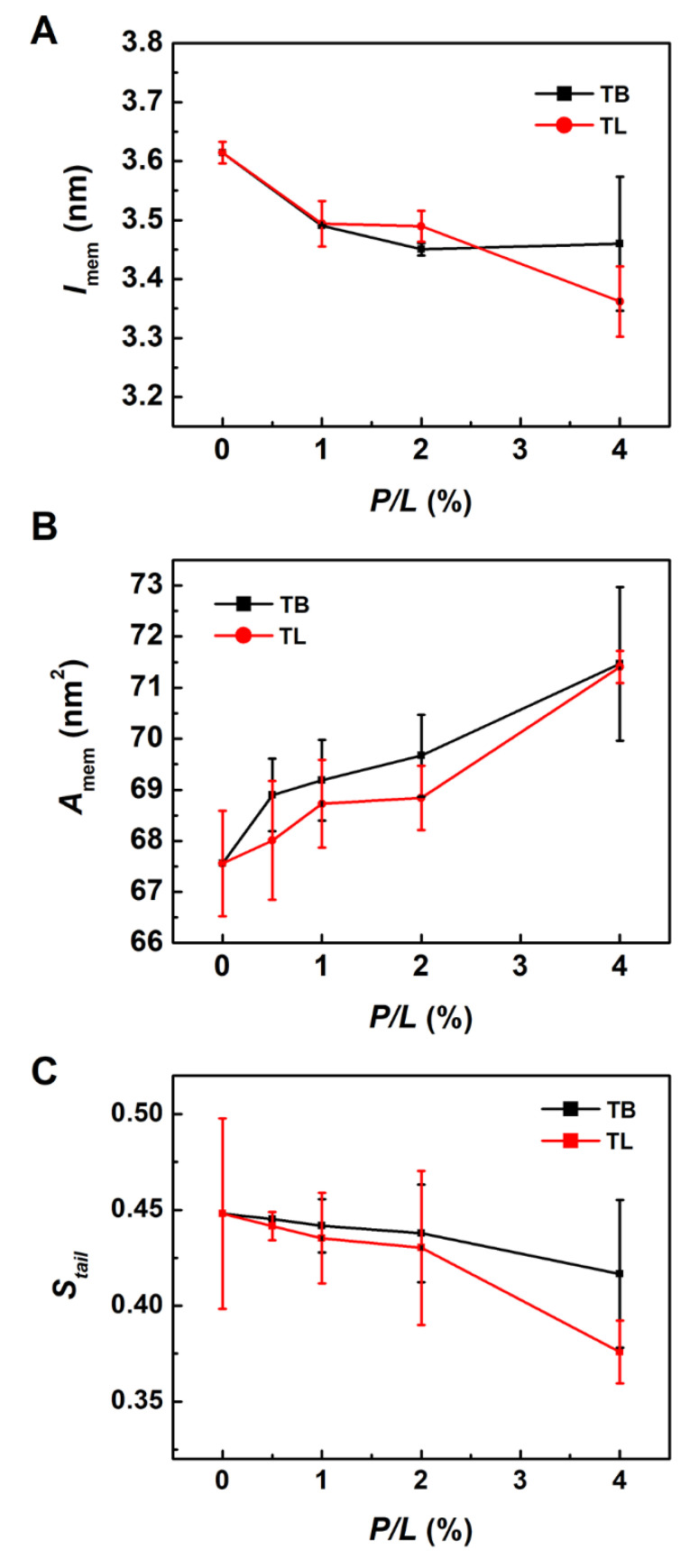
Changes of (**A**) membrane thickness *l*_mem_, (**B**) membrane area *A*_mem_ and (**C**) lipid orientation order induced by TB and TL at various peptide concentrations. Data were obtained from AAMD simulations.

**Figure 5 ijms-22-11015-f005:**
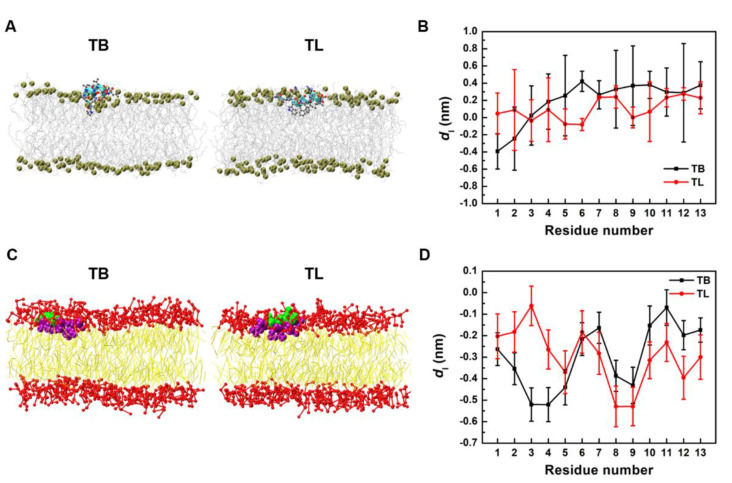
(**A**) Snapshots of binding states of TB and TL monomers on membranes obtained from AA simulations. (**B**) Insertion depth *d*_I_ of Cα atoms of TB and TL monomers into membrane in AA simulations. (**C**) Snapshots of binding states of TB and TL monomers on membranes obtained from CG simulations. (**D**) Penetration depth of backbone beads of TB and TL monomer into membrane in CG simulations.

**Figure 6 ijms-22-11015-f006:**
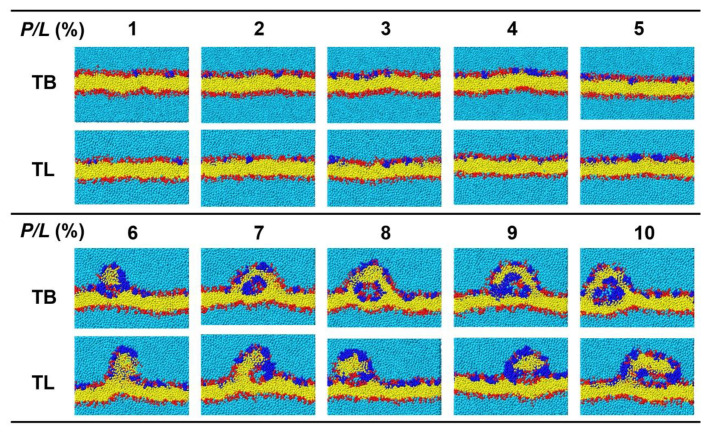
Cross-sectional snapshots of lipid bilayer bound by TBs and TLs at a variety of peptide concentrations. Blue, red, yellow, and light blue beads represent peptides, lipid head groups, lipid tails and water, respectively.

**Figure 7 ijms-22-11015-f007:**
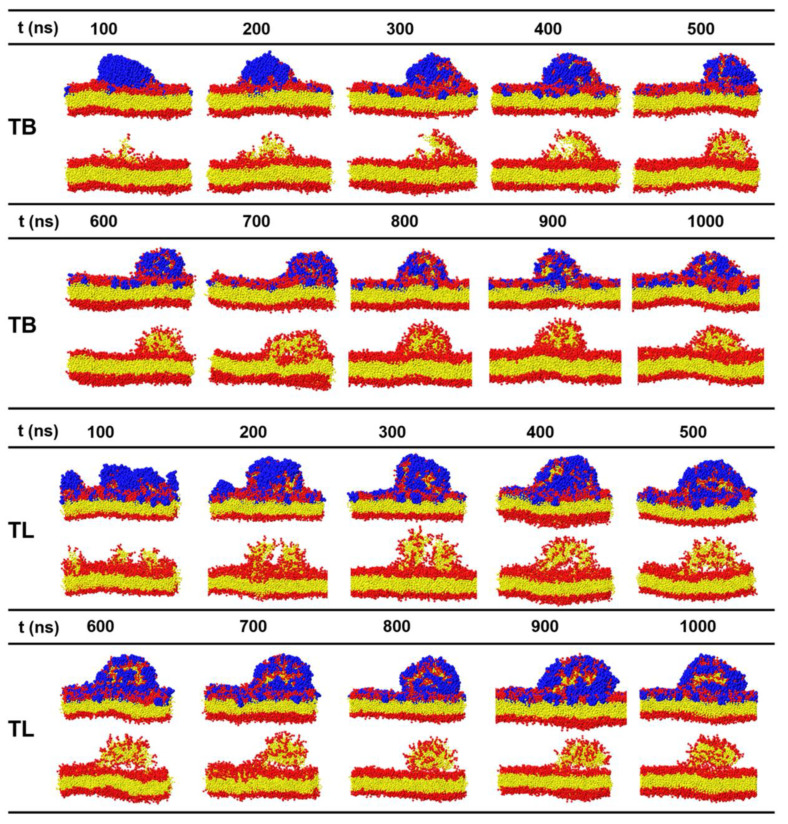
Time sequences of snapshots of lipid bilayer treated with TBs and TLs at a concentration of *P/L* = 10%.

**Figure 8 ijms-22-11015-f008:**
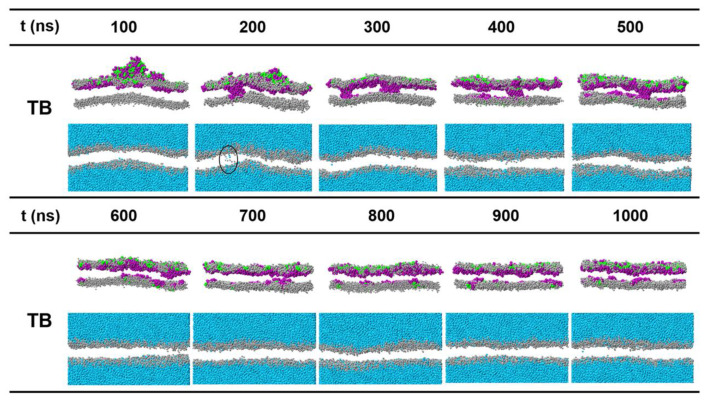
Time sequences of snapshots of planar bilayers treated with temporin B at a concentration of *P/L* = 7.3%. Lipid tails are not shown for clarity. In the second line, water and lipid heads are shown separately. As circled, a few water beads can pass through the channel.

**Figure 9 ijms-22-11015-f009:**
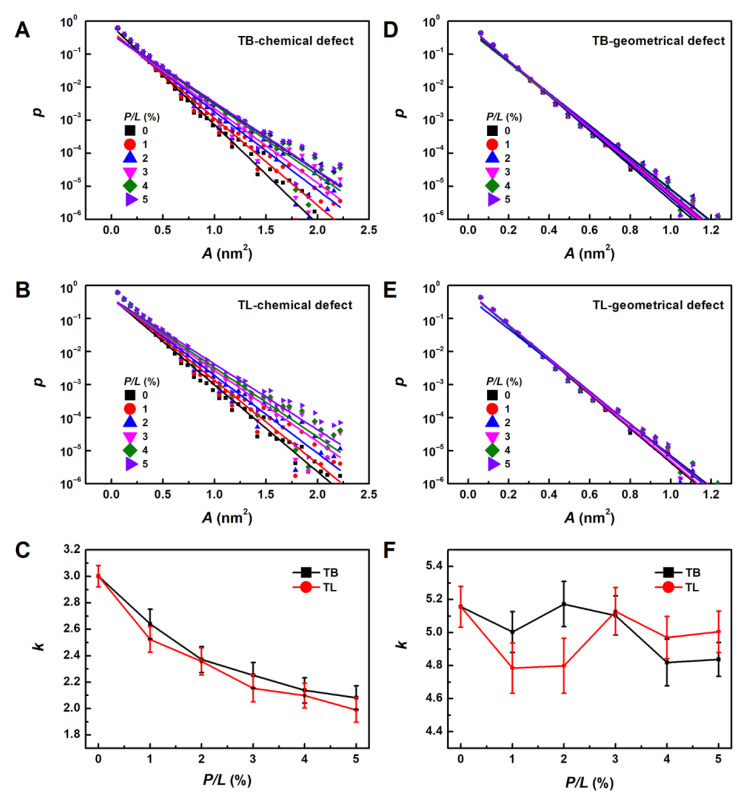
Emerging probability of chemical lipid packing defect as a function of defect area induced by TBs (**A**) and TLs (**B**). (**C**) Exponential decay rates of chemical defect at various peptide concentration. (**D**–**F**) correspond to geometrical lipid packing defects.

**Figure 10 ijms-22-11015-f010:**
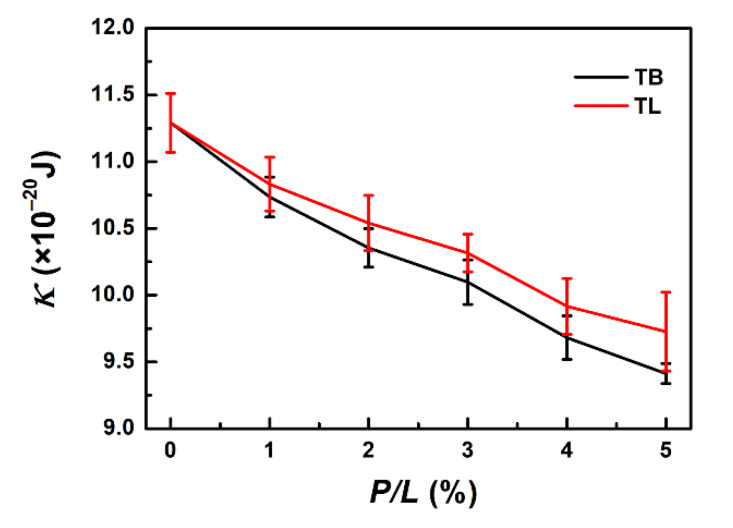
Bending rigidities of TB- and TL-bound bilayer membranes at various peptide concentrations.

## Data Availability

Data related to this article are available from the authors upon reasonable request.

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
