# Peer review of "Structure and Formation Mechanism of Antimicrobial Peptides Temporin B- and L-Induced Tubular Membrane Protrusion"

_ijms, 2021, doi:10.3390/ijms222011015_

Round 1
Reviewer 1 Report
Authors studied the bactericidal mechanism of temporins, a family of antimicrobials peptides, by using all-atom and coarse-grained molecular dynamics. I think this is a very interesting work, since elucidating the mechanisms of membrane damage can help rationalize the design of new, high-effective, peptide-based antibiotics. Although I consider this work as suitable for IJMS, I would like to discuss first with the authors some issues of the computational design and the results that they obtained.
- Authors described in the Methodology of the potential of mean force that each window was simulated until 30 ns. I have doubts about this time length being enough to sample the conformational ensemble of the peptide, principally if conformational changes occur. Thus, I would like to ask the authors if they can provide some results of the convergence tests that prove that this time is sufficient. Moreover, I think they did not specify the time size window for the CG simulations. On the other hand, I don't think this analysis is fundamental for the conclusions of their work. Simulations show the difficulties of the monomer peptides (at low concentration) to penetrate the membrane, which I already consider a good qualitative result.
2. I missed in the Discussion section a more extended comparison of the different mechanisms proposed in the bibliography with the results of this work. For example, it is possible that "the red carpet" mechanism could only occur at low peptide concentrations.
3. Authors claimed in page 11, line 358 that "Compared with TB, TL has one more electric charge which permits it to bind more closely with negatively charged POPG. Meanwhile, the three aromatic residues of TL are more likely to contact with the lipid alkane groups. These two features enable TL to create relatively larger chemical defects, which may explain its cytotoxicity." I am afraid I don't see any significant difference between the TB and TL results in Fig. 9A,B. Indeed, I wonder about the quality of these curve fittings. It is really difficult to distinguish which points correspond to a certain concentration, but it seems that at certain concentrations the behavior is even not linear. In my opinion, authors should show clearly that TL creates larger chemical defects as they claimed.
4. I am certainly interested in the formation of peptide aggregates at the membrane surface that one can see at 10% concentration in Fig 7. I would like to ask the authors if they already started the simulation from an aggregate formation or if it occurred during the first 100 ns. Did they study the growing rate, in that case?
5. Since I am not familiar with the experiments of bactericidal peptides, I wonder if the concentrations that authors considered in this work have any relation with experimental designs (both in vitro and in vivo). I think the reader would appreciate this clarification to understand, for example, if concentrations of 10% are just standard in antibiotics or this corresponds to really high concentrations.
Reviewer 2 Report
The manuscript “Structure and Formation Mechanism of Antimicrobial Peptides Temporin B and L Induced Tubular Membrane Protrusion” submitted by Shan Zhang and coworkers reports a comprehensive molecular dynamics study on antimicrobial peptides. In particular, the authors used all‐atom and coarse‐grained molecular dynamics simulations to study the structure and interactions with model membranes of these peptides. Overall, the manuscript is well organized, and the topic covered is interesting for the field. In my opinion, additional data and discussions are needed.
Major and minor comments:
- The authors used a mixture of zwitterionic lipids (POPC and POPG in a 7:3 molar ratio) to investigate the interaction of the antimicrobial peptides with the bacterial plasma membrane. However, other minoritarian lipids are present in certain bacterial membranes such as cardiolipin (CL). Please discuss how these minoritarian lipids may influence the behavior of TB and TL.
- All the molecular dynamics were performed with antimicrobial peptides (TB and TL). Here, the results clearly demonstrate the strong effect of these peptides on the membranes. However, I would suggest the authors to evaluate the behavior of control peptides with a equivalent size and properties but without antimicrobial properties for comparative purposes. This new results should be presented and discussed in the manuscript.
- The authors claim a concentration-based behavior of the peptides. Please discuss if the modelled concentrations are compatible with the physiologic concentrations of these peptides that are found in nature.
Round 2
Reviewer 1 Report
Authors adequately addressed all my comments in the revised version of the manuscript.
Reviewer 2 Report
The authors added additional discussion in the revised version of the manuscript. Now the manuscript is suitable for publication in present form.